# Environmental change makes robust ecological networks fragile

Giovanni Strona[1] & Kevin D. Lafferty[2]

Complex ecological networks appear robust to primary extinctions, possibly due to consumers' tendency to specialize on dependable (available and persistent) resources. However, modifications to the conditions under which the network has evolved might alter resource dependability. Here, we ask whether adaptation to historical conditions can increase community robustness, and whether such robustness can protect communities from collapse when conditions change. Using artificial life simulations, we first evolved digital consumer-resource networks that we subsequently subjected to rapid environmental change. We then investigated how empirical host–parasite networks would respond to historical, random and expected extinction sequences. In both the cases, networks were far more robust to historical conditions than new ones, suggesting that new environmental challenges, as expected under global change, might collapse otherwise robust natural ecosystems.

[1] European Commission, Joint Research Centre, Institute for Environment and Sustainability, Via E. Fermi 2749, 21027 Ispra, Italy. [2] US Geological Survey, Western Ecological Research Center c/o Marine Science Institute, University of California, Santa Barbara, California 93106, USA. Correspondence and requests for materials should be addressed to G.S. (email: goblinshrimp@gmail.com).

Ecosystems, whether coral reefs, rain forests or a microbiome, are complex. How complexity evolves and its robustness to change is both a mystery to ecologists and a challenge for conservation biologists. To assess the current biodiversity crisis during global change requires looking beyond endangered species lists to extinction cascades[1]. Secondary extinction risk should be highest for specialists, an example being the loss of the condor louse after the condor became extinct in the wild[2]. In some circumstances, secondary extinctions could trigger even more extinctions up a food chain, unravelling entire ecosystems[3]. For this reason, the already appreciable endangered species list might be just the beginning[4].

Although evolution should lead to both specialization and robustness to species loss, global change and human activity might change species vulnerabilities[5,6], which could then decrease resource–consumer network stability. We investigated this hypothesis by contrasting how historical and novel conditions affected parasite assemblage robustness using computer simulations and information from global host–parasite databases. Ecological networks were fragile under environmental change due to the tradeoff between adapting to a stable past or an uncertain future.

Parasite assemblages are a convenient model for studying network robustness because they provide a straightforward, unidirectional response to host species loss (that is, host extinction, in general, affects parasite persistence but not the other way around). To estimate parasite assemblage robustness, one can apply analytical models, but it is more accurate to take a food web or a bipartite host–parasite network and then remove hosts in sequence (called host disassembly), recording the rate at which parasite richness declines[7]. Parasite robustness declines

faster with increasing life-cycle complexity and parasite specificity[7]. However, parasite specificity is not independent from host extinction order. For instance, fish parasites tend to be either generalists or they specialize on 'dependable' hosts that are less vulnerable to extinction[8]. A similar pattern occurs within local food webs, with host/parasite networks being more robust to rare host removal than to random host removal[7]. However, current host vulnerability to extinction, measured by modern threats (for example, habitat destruction), can differ from historical vulnerability to extinction[9], suggesting that dependable hosts in the past might not be dependable in the future[5,6,10].

Here, we use artificial life simulations and empirical data to investigate how stable ecosystems would respond to extinctions under different scenarios of species loss. We show that ecosystems evolve complexity that is robust to historical conditions. However, under changing conditions, including current anthropogenic threats to biodiversity, robustness to change decreases, suggesting that future species losses should trigger secondary extinctions and eventual ecosystem collapse.

## Results

**Evolving digital host–parasite networks.** We ran robustness experiments with digital hosts and parasites that self-replicate, mutate and compete on the artificial life platform Avida[11]. We chose different random settings covering various environmental scenarios (see the 'Methods' section and Supplementary Table 1) for each of the 100 evolution simulations. Each simulation started from a single host ancestor that 'speciated' with time. After a random number of host generations, we injected a random number of identical parasite individuals. As hosts and parasites diversified, host species varied in their vulnerability to extinction, and parasite species varied in their virulence, that is, in the percentage of central processing unit (CPU) cycles subtracted from a host. Because carrying capacity (that is, number of available hosts) remained constant throughout the simulation, the increased host and parasite diversity resulted in an average increment in specialization of interactions (Fig. 1a). Although specialization is expected to increase co-extinction risk, parasite assemblages gained robustness over time, reaching a maximum within about $5 \times 10^4$ generations (Fig. 1b). We continued the simulations until $1–5 \times 10^5$ generations, obtaining robust digital host–parasite interaction networks at different stages of maturity.

After each co-evolution simulation, we assessed host vulnerability (measured as the fraction of simulation steps survived by a host) by halting mutation and then letting organisms reproduce and interact with the other hosts and parasites they had evolved with. Competitive interactions, along with stochastic processes, extirpated host species in sequence (like in an 'arena' experiment) until the least vulnerable host remained[12].

Host population size at the end of the co-evolutionary phase was a good predictor of a host's vulnerability to extinction (with average Spearman rank correlation coefficient between number of starting individuals and survived steps in the no-mutation phase, $r_s = 0.81$, $P < 0.01$ in 96% of cases). Vulnerable hosts had parasites with broader host ranges in most co-evolved networks (average Spearman rank correlation coefficient, $r_s = 0.54 \pm 0.21$, $P < 0.05$ in 78% of cases), which shows that, over time, parasites only specialized on dependable hosts. Furthermore, in most networks, we also found a negative relationship between the number of parasite species able to infect a host species and host vulnerability to extinction (with average $r_s = -0.68 \pm 0.15$, $P < 0.05$ in 91% of cases). Thus, the most dependable hosts had the highest parasite diversity and allowed for host specialization, whereas

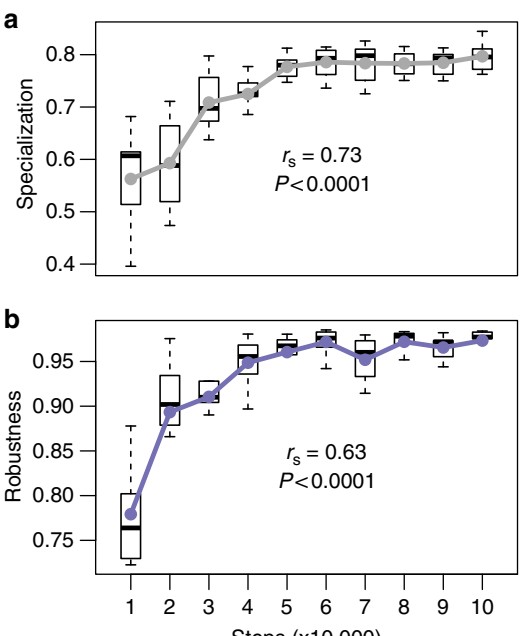

**Figure 1 | Robustness evolves rapidly in Avida even while specialization increases.** Robustness (**a**) was measured as the area under the curve of parasite diversity versus host diversity when a network was disassembled according to host historical vulnerability (see the 'Methods' section). Specialization (**b**) was computed as 1 minus the average fraction of hosts used by parasites in a given experiment. Solid lines and dots indicate average values. The Spearman-rank correlation coefficient was computed on all raw data (that is, not on the averaged values shown in the plots). Boxes indicate first and third quartiles, whiskers indicate range values and horizontal lines indicate median values.

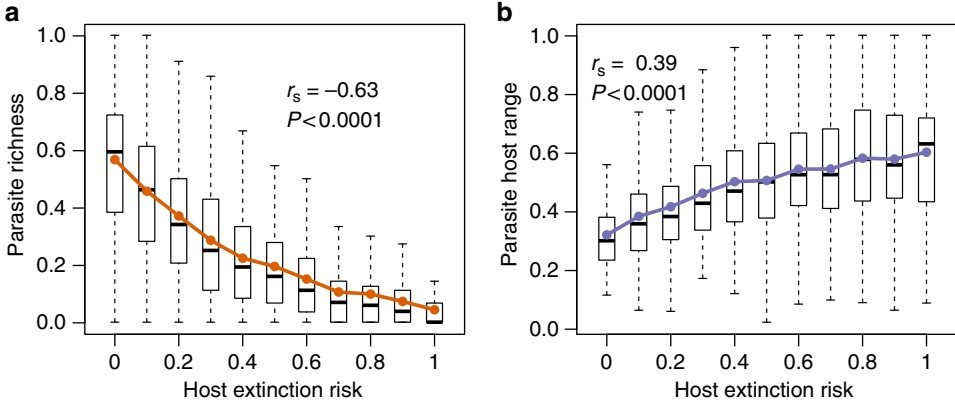

**Figure 2 | Vulnerable hosts in Avida have fewer and more generalist parasites.** Host extinction risk was measured as $h/H$, with $h$ being the ordinal position in the extinction sequence of the target host as assessed after the coevolution phase, and $H$ being host richness. Parasite richness (**a**) and host range (**b**) were measured, respectively, as the average fraction of parasites infecting a host, and of hosts used by a parasite. Solid lines and dots indicate average values. The Spearman-rank correlation coefficient was computed on all raw data (that is, not on the averaged values shown in the plots). Boxes indicate first and third quartiles, whiskers indicate range values and horizontal lines indicate median values.

only generalists could afford to parasitize undependable hosts (Fig. 2). This helps explain how assemblage robustness and specialization can increase together over time.

**Disassembling ecological networks**. To measure parasite assemblage robustness to secondary extinction, we disassembled the 100 host–parasite networks by removing hosts in sequence until all hosts were removed and all parasites suffered secondary extinction. We did this under four sequence treatments: best case, worst case, historical and random (Supplementary Movie 1). The 'best case' scenario removed hosts in the reverse order of parasite richness, thereby maximizing robustness. In other words, because hosts harbouring many parasites stayed in the assemblage longer than hosts having lower parasite richness, parasite richness declined slowly with host removal. Similarly, we found the lowest robustness (the 'worst-case' scenario) by removing hosts in the order of parasite richness, causing parasite richness to decline rapidly with host removal. With the robustness limits established, we removed hosts from most to least vulnerable (according to the extinction sequence observed in our assessment phase) to investigate robustness to historical conditions. Then, we simulated novel changes to the system by randomizing host removal order. By using randomization to mimic environmental change in Avida, we do not mean to imply that climate change or human activity randomly affects species vulnerability, just that relative species vulnerabilities might differ from historical conditions in unpredictable ways, as seen for fishes[9].

To compare parasite assemblage robustness among disassembly treatments, we plotted the points representing the fraction of extant parasites after each host removal versus the corresponding fraction of extant hosts, and we quantified robustness as the area under the curve connecting these points. The results were not affected by genotypes, network structure, initial conditions or whether we removed hosts on the basis of their functional complexity (see the 'Methods' section). Parasite assemblage robustness under historical vulnerability was close to the best-case scenario, showing that parasites had evolved to maximize robustness by not specializing on vulnerable hosts (Fig. 3, Supplementary Movie 1). Conversely, parasite assemblage robustness under random host vulnerability (that is, our proxy for environmental change), was intermediate between the best- and worst-case scenarios (Fig. 3, Supplementary Movie 1), showing that changing conditions reduced robustness (an effect that strengthened with time spent in the historical condition,

Fig. 4). The more historically stable the host assemblage, the less robust the parasite assemblage was to random host removal.

**Experiments on empirical networks**. To test our hypothesis with actual hosts and parasites, we compiled networks including all known helminth parasites from vertebrate hosts. The global host–parasite databases have different properties from Avida. Most notably, the global networks are incomplete, they include species that, due to different geographical distributions, do not co-occur and interact, and many parasites have complex life cycles (in the 'Methods' section, we discuss why these limitations do not alter the main findings). Regardless, as in Avida, the host–parasite associations in global databases should, in principle, reflect the relationships between resource dependability, and parasite richness and specificity. Fish parasite data were from FishPest[13], whereas data on amphibians, birds, mammals and reptiles and their acanthocephalans, cestodes, nematodes and trematodes were from the Natural History Museum of London host–parasite database (www.nhm.ac.uk), aggregated as in Strona and Fattorini[14].

One advantage to empirical data over artificial life simulations is that we could infer novel host vulnerability to extinction from International Union for Conservation of Nature (IUCN) risk categories (www.iucnredlist.org) in addition to removing hosts at random. From these data, we simulated novel, best-case and worst-case host-removal scenarios. Although we did not know historical vulnerability for the terrestrial vertebrates, we used fish intrinsic vulnerability[9,15] to represent historical vulnerability in fishes[8].

Although removing fish according to their intrinsic vulnerability to extinction (corresponding to the historical conditions in our digital experiments) resulted in significantly lower than best-case parasite assemblage robustness, robustness was still higher than when removing fish under novel conditions, whether these novel conditions were based on IUCN risk categories (that is, future vulnerability) or random removal (Fig. 5). For both fish and the other host groups where historical vulnerability was not available, robustness under novel vulnerability to extinction measured by IUCN risk categories was no different from random host removal (paired $t$-test on the respective areas under the curve = 0.48; Fig. 6). Furthermore, the robustness to novel change was indistinguishable among different host and parasite taxa (analysis of variance $P = 0.53$ for host taxa and $P = 0.6$ for parasite taxa), suggesting these results are general.

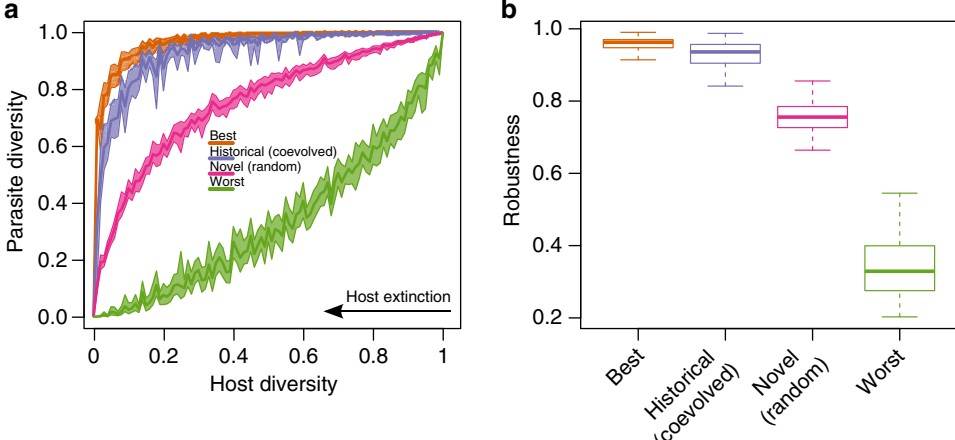

**Figure 3 | Robustness in Avida is close to the best-case scenario under historical conditions but declines under novel conditions.** (**a**) Average fraction of parasite species remaining in the host–parasite network after the subsequent removal of all host species according to historical conditions (from right to left) in all the experiments, contrasted with a best and a worst case scenario, and a novel (random) scenario. Ribbons along solid lines indicate 95% bootstrapped confidence intervals. (**b**) Robustness (the areas under curves in the four host removal scenarios). Boxes indicate first and third quartiles, whiskers indicate range values and horizontal lines indicate median values.

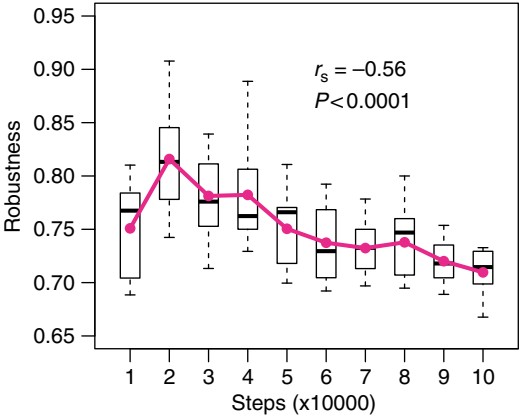

**Figure 4 | Robustness to novel conditions in Avida declines over time due to a tradeoff with the adaptation to stable conditions.** Solid lines and dots indicate average values. The Spearman-rank correlation coefficient was computed on all raw data (that is, not on the averaged values shown in the plots). Boxes indicate first and third quartiles, whiskers indicate range values and horizontal lines indicate median values.

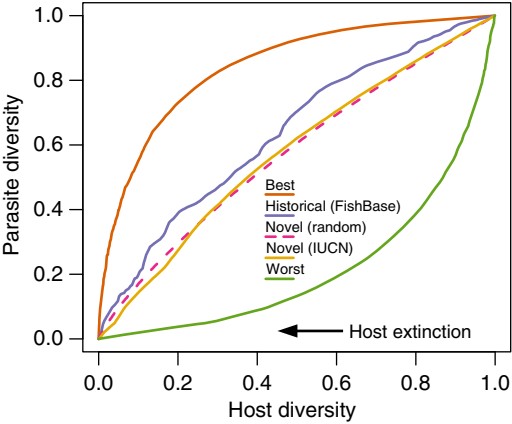

**Figure 5 | Fish parasites are more robust to historical conditions than to novel ones.** Extinction trajectories in a network based on the entire FishPEST database[13] disassembled, according to a best-case and a worst-case scenario, historical conditions based on intrinsic vulnerability values[15], and two novel scenarios (IUCN extinction risk categories and random removal).

## Discussion

Past studies found that historical and novel conditions affect network robustness. For instance, empirical networks evolve non-random structure that improves robustness to current conditions[16]. As in our historical disassembly experiments, lake food webs are more robust to removing the most vulnerable species than the least vulnerable species, a result attributed to specialists being more likely to feed on invulnerable prey[17]. Likewise, in a salt marsh food web, the parasite assemblage is more robust to rare host removal than to random host removal[7]. In contrast, the low parasite assemblage robustness to removal based on host vulnerability in the Upper Paraná River floodplain is likely due to several fish species introductions[18], which exemplifies how global change might reduce network robustness by altering historically assembled networks. In such cases, robustness to change might further decrease with specialization, as observed in digital networks (Supplementary Figure 1).

These model systems have their limits. In natural systems, network robustness varies with spatial and temporal scales. At small spatial scales, robustness depends on patch dynamics,

and colonization might ameliorate extirpations, whereas at the global scale, robustness depends on extinction, and effects are more long-lasting, at least until evolution can replace lost species. Because the Avida experiments stop mutation after the assessment phase, the results do not apply to evolutionary time scales, and, because they do not allow colonization from outside, they do not apply to patch dynamics. Furthermore, because the global empirical host–parasite disassembly experiment includes hosts and parasites that do not overlap in habitat or space, it does not mimic assemblage dynamics. Despite these different assumptions, the results were similar. In the future, we hope to test our hypotheses in empirical host–parasite networks measured within a particular ecosystem.

Parasites can indicate rich, complex, healthy ecosystems because more host species create opportunities for more parasite species to establish[19–21]. We see this in our 100 Avidian host–parasite assemblages, where high parasite diversity reflects high host diversity ($r_s = 0.62$, $P < 0.005$). In addition to confirming how parasite richness indicates host richness, we can now suggest a new use for parasites as indicators. Because assemblage

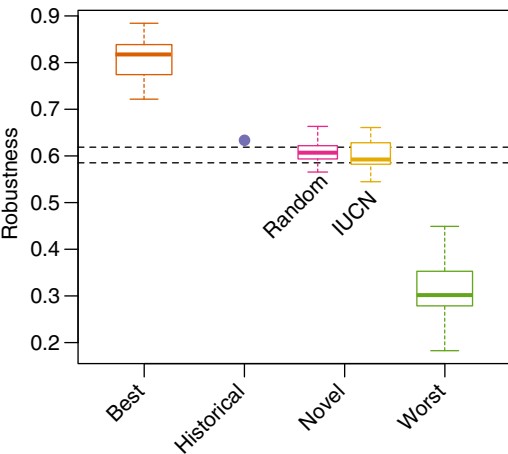

**Figure 6 | The similar low robustness of global host-parasite networks to current conditions and random ones suggests they are not adapted to global change.** Robustness represents the average parasite assemblage robustness for the global fish parasite network and 16 large host–parasite networks for terrestrial vertebrates (amphibians, birds, mammals and reptiles versus acanthocephalans, cestodes, nematodes and trematodes) according to a best-case and a worst-case scenario, and two novel scenarios (IUCN extinction risk categories and random removal). The blue dot indicates the robustness of the global fish network when disassembled according to FishBase vulnerability values (area under the blue solid line in Fig. 5). The dotted lines indicate upper and lower 95% confidence intervals for the robustness values in the random novel scenario. Boxes indicate first and third quartiles, whiskers indicate range values and horizontal lines indicate median values.

robustness to novel change declines with time spent in stable environmental conditions (Fig. 4), parasite assemblage robustness to random host removal could be used to assess past host assemblage instability, a community property that is difficult to measure directly. Furthermore, because parasites tend to specialize under stable conditions (Fig. 1a, Supplementary Fig. 1), a simple inverse measure of historical environmental stability is the average number of hosts per parasite. By combining average number of hosts per parasite and parasite richness, one could estimate whether host communities are relatively rich and stable, rich and unstable, poor and stable, or poor and unstable. Then, if conditions change, we predict most secondary extinctions should occur in rich and stable systems, such as rainforests and coral reefs

Whether *in silico*, or in nature, ecological networks should assemble over time so that consumers avoid specializing on vulnerable resources. Ironically, this stabilizing mechanism promotes specialization, which then decreases network robustness to novel conditions as expected under climate change, species invasions and habitat alteration (Supplementary Fig. 1). Although there will be winners and losers with global change[22], our findings suggest that winners cannot balance losers when it comes to network robustness.

## Methods

***In silico* experiments.** We ran digital experiments on the artificial life platform Avida version 2.14.0 (http://avida.devosoft.org/)[11]. The Avida platform is a model system for investigating how co-evolution (and co-extinction) works within simple, transparent, stochastic rules[11,23]. The most important difference between empirical food-web assembly[24,25] and artificial life simulations is that food-web biologists model natural-looking systems to explain patterns in nature, whereas artificial-life simulations are alternative 'natural' systems that evolve *in silico*[26,27]. Artificial life simulations generate and maintain complexity starting from a few rules (just as in natural evolution). This, in turn, generates the patterns expected in ecological and evolutionary systems.

We studied host–parasite networks for several reasons. First, although Avida can model predator–prey interactions[28], and can create different trophic levels by manipulating the resource setting[29], these options are unexplored. Conversely, host–parasite networks in Avida have ecological and evolutionary behaviors close to those observed in nature[28], and have been already used to explain how complex features evolve[30] and how parasites maintain host diversity[12]. Furthermore, empirical food webs have limitations, such as unequal resolution among the trophic levels due to lumping some species in broad taxonomic categories[31]. For instance, in aquatic trophic webs, it is common to have fish identified at the species level, and all the phytoplankton aggregated into a single category[32]. This creates obvious problems for interpreting robustness, especially when lower trophic levels are aggregated, leading to unrealistic assumptions such as all phytoplankton go extinct at once as if they were a single species. Additionally, there is little information from IUCN on extinction risk for non-vertebrate species. Thus, even if we obtained resolved food webs, we would have not been able to investigate their robustness to future extinction scenarios. More importantly, host–parasite networks are bipartite, which makes it easy to isolate how secondary extinctions affect system robustness, whereas in food webs, hosts can suffer secondary extinctions if their resources vanish[7].

We used the Python programming language[33] and R[34] to process Avida output, simulate disassembly, and analyse data. Supplementary Data 1 details the co-evolved host–parasite networks and the host extinction sequences in the historical scenario.

The Avida study had three distinct phases: coevolution, assessment and disassembly. In the co-evolution phase, we ran simulations to evolve 100 complex host–parasite networks (100 being enough for hypothesis testing in past studies[23,30]). To make our results as general as possible, we randomized several parameters (carrying capacity, parasite virulence, resource availability, injection timing and amount of ancestral parasites) of a setting already explored in host–parasite co-evolution experiments[30]. In each simulation, the Avida world was a bi-dimensional grid with random dimensions between 50 and 120 host units (thus between 250 and 14,400 host habitations). Mutation rates were the same as in Zaman *et al.*[30], but we also allowed parasite virulence to mutate throughout the experiment (starting from 1, that is, a situation in which the parasites steal all the CPU to its host). To create environmental variation, we randomly selected between one and nine resources associated with the canonical nine Avida logical operations (*not*, *nand*, *and*, *orn*, *or*, *andn*, *nor*, *xor* and *equ*). Similarly, we randomly associated the available resources as products of the nine tasks, with an input–output ratio randomly selected between 0 and 0.5. We set a random seed for each replicate, inserted a single host ancestor and allowed for host diversification for a random period lasting between 1,000 and 5,000 steps, at which point we injected 500–1,000 individuals belonging to a single ancestral parasite species. Both the host and the parasite ancestors could only do one of the two least complex tasks in Avida (that is, the 'NOT' function, where 0 is returned if 1 is consumed, and vice versa)[30]. Parasites in Avida are similar to free-living species in terms of structure and evolutionary processes (that is, mutation type and rate). However, parasites could not survive outside a host. Thus, when a parasite reproduces, its offspring try to infect a nearby host (like a directly transmitted, single-host microparasite). The host is susceptible only if it is uninfected, and the parasite can do at least one of its tasks. Depending on their virulence, parasites can take up to all the host's CPU cycles. During the co-evolution phase, hosts and parasites competed, interacted and co-evolved, generating complex host–parasite networks with different structural properties. We retained the first 100 simulations where at least one host and parasite species persisted to a random end point between $10^5$ and $5 \times 10^5$ steps (resulting in assemblages with different ages).

After stopping the co-evolution phase, we assessed host vulnerability to extinction by letting host species interact but not mutate[12]. In this context, host species compete, going extinct one after another, depending on their relative fitness, providing an objective way to measure host vulnerability to extinction under the conditions in which they evolved in the coevolution phase (that is, historical conditions).

In both the co-evolution and assessment phases, we monitored the species every 100 steps by recording genome, spatial position, host or parasite, genetic code and closest ancestor. The Avida documentation at https://github.com/devosoft/avida/wiki gives additional details about how to setup/run Avida simulations, and processing/interpreting Avida output.

In the assessment phase (that is, after we stopped mutations), we also recorded, for each host: (i) vulnerability to extinction, measured as $h/H$, with $h$ being the order a species went extinct and $H$ being the starting host abundance; (ii) parasite richness; and (iii) average parasite host range; that is, the average number of hosts (including the target one) used by its parasites. Then, we assessed the relationships between parasite richness and average host range (standardized by, respectively, the total number of parasite species and the total number of host species per simulation), and host vulnerability to extinction (using Spearman's rank correlation coefficient) on each individual run, and on all the runs aggregated.

In the disassembly phase, we removed hosts one by one until no hosts remained, counting parasite species richness at each step. We determined the best-case scenario for parasites by removing, in sequence, the host with the least parasites, so that hosts with many parasites went extinct late and parasite diversity declined slowly. We determined the worst-case scenario for parasites by removing, in sequence, the host with the most parasites, so that hosts with many parasites

went extinct early, and parasite diversity declined rapidly. Then, to identify the parasite assemblage robustness to historical conditions, we removed, in sequence, the most vulnerable host to extinction as quantified in the assessment phase. Finally, to identify parasite assemblage robustness to a hypothetical environmental change, we randomized the host-removal sequence.

For each removal treatment, we averaged 100 replicates where we randomized ties (for example, the host extinction order associated with the same number of parasites in the best- and worst-case scenario, or having the same historical vulnerability or the same maximum complexity). In the random scenario, we simply averaged 100 random sequences. We quantified parasite assemblage robustness as the area under the host diversity versus parasite diversity curve (rescaled as proportions)[7].

We considered the extent that various assumptions might affect our results. Specifically, we asked how network structure, task complexity, genotype, incomplete information and time in the co-evolutionary phase affected robustness. As detailed in the following paragraphs, these factors did not alter the qualitative findings.

Because robustness can be sensitive to network structure[35], we controlled for network structure by applying each treatment to the same network. Nonetheless, our replicate networks differed in structure and that could lead to differences in robustness among them, adding to the variation we see in robustness within a treatment. To investigate whether and how network structure can affect robustness, we did pairwise comparisons between robustness, and basic network properties such as the number of nodes and edges, the connectance, nestedness/overlap and modularity[36]. Most networks tended toward either a nested or a modular structure or both (Supplementary Fig. 2), whereas segregation (that is, the tendency of species to minimize overlap in partners) never emerged, which might support the idea that sharing interacting partners could promote network persistence. Usually, network structure (Supplementary Table 1) did not affect robustness, explaining, at most, 10% of the variation (Supplementary Table 2).

We investigated whether task complexity (see Table 1 in Lenski et al.[23]) affected parasite assemblage robustness by replicating the disassembly experiments on the 100 networks and removing hosts in decreasing or increasing order of complexity. We measured host complexity as equal to its most complex task. Complexity did not correlate with robustness ($P = 0.21$). Moreover, the robustness measured using decreasing complexity was not distinguishable from that obtained from random removal ($P = 0.56$), whereas robustness measured using increasing complexity was only slightly different from the random removal ($P = 0.047$; see also Supplementary Fig. 3).

To identify and disassemble networks, we classified organisms (both host and parasites) into taxonomic units (hereafter 'species') by their phenotype, that is, their ability to do particular tasks[30]. However, to assess whether evolutionary convergence affected the results, we replicated all the disassembly experiments by classifying taxonomic units by genotypes. We found consistent patterns suggesting that evolutionary convergence was not biasing our findings (Supplementary Fig. 4).

Empirical networks suffer from incomplete information that underestimates generality (for example, the more we study networks, the more complex they appear, with increasing redundancies). To better compare the simulations with the analysis on the (undersampled) empirical networks, we asked what simulations would have looked like if we had incomplete information. We replicated all the analyses by eliminating 10, 20, 30, 40, 50, 60% of the host–parasite interactions, finding that data gaps only slightly underestimated parasite assemblage robustness (Supplementary Figs 5–8).

To investigate how soon parasite assemblage robustness emerges through the co-evolutionary phase, we measured a random parasite assemblage's robustness to historical and novel perturbations every 1,000 steps, from 10,000 to 100,000 generations. Figures 1 and 4 show that robustness to historical conditions evolved relatively early, but there is a tradeoff with robustness to novel conditions.

**Tests on empirical host–parasite networks.** As a companion to the Avida experiments, we evaluated our predictions using all fish–parasite records available from FishPEST (http://purl.oclc.org/fishpest)[13], and 16 large host–parasite networks, built by combining all the records available from the host–parasite database of the Natural History Museum of London (http://www.nhm.ac.uk/) for different combinations of host and parasite taxa (amphibians, birds, mammals and reptiles versus acanthocephalans, cestodes, nematodes and trematodes). These networks have already been used to estimate global parasite richness[14].

Both datasets were filtered by including only hosts present in IUCN redlist. After this filtering procedure, the FishPEST network included 16,681 associations between 1,696 fish species and 7,555 parasite species belonging to different taxa (acanthocephalans, cestodes, monogeneans, nematodes, trematodes), whereas the Natural History Museum of London data included 29,275 associations between 2,751 host species and 9,638 parasites species.

We disassembled these networks, by taxon, by removing hosts as in our experiments on digital networks. For all host groups, we could estimate future vulnerability from IUCN risk categories for some species. Thus, for both the vertebrate and the fish–parasite networks, we simulated 100 worst-case, best-case and novel scenarios (randomizing the whole order of extinctions in the random scenario, and only ties in the others). For the fish–parasite network only, we simulated 100 disassembles under historical conditions using fish intrinsic vulnerability to extinction (www.fishbase.org)[15] as a proxy for historical vulnerability. Such measures, which take into account fish life history and

ecological characteristics, have already been used as a proxy for fish intrinsic extinction risk in studies dealing with co-extinctions[8,18].

**Sensitivity analyses.** We considered the extent that various assumptions might affect our empirical results. For instance, we calculated robustness as if all parasite species had simple life cycles. However, many parasites require two or more host species in sequence, and this reduces robustness[7]. Another aspect that reduces robustness in food webs (absent from our disassembly) is that hosts can suffer secondary extinctions if their resources vanish[7]. However, because the overestimation applies to all the scenarios, we expect no bias in the results. The following paragraphs describe how we determined that reducing the FishPest data filtering with IUCN records, ties in the IUCN vulnerability rankings, incomplete parasite information and sampling-effort bias did not alter the qualitative findings.

Although filtering the FishPEST data set by IUCN records reduced the data set, our analysis on Avida partial networks suggested that incomplete information does not introduce biases, instead underestimating parasite assemblage robustness. We found the same pattern by replicating the fish host removal according to intrinsic host vulnerability on the complete FishPEST network (including 33,426 unique interactions between 12,762 parasite species and 4,091 host species). Supplementary Figure 9 plots parasite diversity versus host diversity for, respectively, the complete FishPEST network, and that filtered by IUCN data (that is, the same curve shown in Fig. 5).

A potential problem with comparing the empirical data is that the IUCN categories lead to more ties in the vulnerability ranking than occurs for the continuous FishBase vulnerability measure[15]. To investigate whether ties affected the results, we replicated the fish host disassembly by grouping fish into five intrinsic vulnerability categories (corresponding to the five intervals in the FishBase vulnerability scale). The resulting historical disassembly curve was almost identical to the one obtained using continuous vulnerability values ($R^2 = 0.99$; regression line: slope = 1.0; intercept = $-0.000004$).

Furthermore, the species that are most under-sampled for parasites are either rare, or limited in range. According to our main hypotheses and findings, rare species are likely to be used by few generalist parasites. Thus, their addition to the data sets would not have a strong effect on parasite persistence. Similarly, missing some host records for widespread generalist parasites would affect the overall robustness little, because these species are already likely to persist. Even if the estimate error was appreciable, we do not expect it to affect the comparison among treatments. The biggest concern about bias for the empirical data would be if increasing host sampling effort disproportionately sampled vulnerable hosts and found more specialist than generalist parasite species. Such a pattern could make it appear that invulnerable hosts had more specialist parasites when they were simply sampled more. Fortunately, sampling effort (measured as published parasite studies per fish species) did not correlate with vulnerability ($r_s = -0.015$) or specialist to generalist ratio ($r_s = -0.049$).

**Data availability.** The 100 co-evolved host–parasite networks and the host extinction sequences in the historical scenario are available in Supplementary Information. Host–parasite records used in the analyses are available from FishPest (http://purl.oclc.org/fishpest) and from the London Natural History Museum host–parasite database (http://www.nhm.ac.uk). Fish vulnerability scores can be obtained from FishBase (http://www.fishbase.org), while conservation status information for both fish and terrestrial vertebrates can be retrieved from the IUCN website (http://www.iucnredlist.org).

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

## Acknowledgements

We thank J. Jokela, D. Morton, S. Fattorini and L. Zaman for useful feedback on a draft manuscript. Any use of trade, product or firm names in this publication is for descriptive purposes only and does not imply endorsement by the U.S. government. The views expressed are purely those of the writers and may not in any circumstance be regarded as stating an official position of the European Commission. This product has been peer reviewed and approved for publication consistent with U.S. Geological Survey Fundamental Science Practices (http://pubs.usgs.gov/circ/1367/).

## Author contributions

Both the authors contributed equally to the paper.

## Additional information

**Competing financial interests:** The authors declare no competing financial interests.

