## [Peer Review File · Nature Communications]

Reviewers' Comments:

Reviewer #1 (Remarks to the Author)

I read with interest the paper by Strona and Lafferty. The simulations using Avida were a very interesting approach to the problem on how coevolution affects the fragility of ecological networks. Having said that I am afraid there are serious issues with the current draft.

First, there is no assessment of the generality of the results.

1. For example, are hundred simulations enough? The authors did not provide any evidence this is a good number of simulations.

2. More importantly, there is no sensitivity analysis on the parameter values used or on modeling decisions (e.g., 400 parasites copies, 2000 generations, and so on).

3. As a consequence - and if I understood right - all the results reported derive from a single combination of parameters and model decisions (which are, of course, implicit parameters of the model). If so, the results are actually preliminary.

4. As a consequence of 1-3 it is impossible to make any assessment of the generality of the results and at it stands the work is a promising set of results, but it is not a solid contribution one would expect in a top journal as Nature Communications.

Second, I could not evaluate the quality of the work presented because I simply could not follow several aspects of the draft. Examples include but are not limited to:

5. I assume the r_s values are the Spearman's rho but if so how it tell us about specialization? Is about how specialization varies with host survival? If so it is not clear. Line 59.

6. Figure 1: how can p-values be smaller than zero?

7. If you are "halting mutation and letting host species outcompete each other in the environmental conditions they had evolved" you are not evaluating the parasitism effect but the consequences of parasites to competition. Is this right?

8. What is the meaning of "parasite robustness"? I do not follow why "removing hosts in decreasing order with parasite richness" is "the best-case scenario for parasite robustness".

9. What is the meaning of "approximated as the average of 100 random orderings"?

10. parasites suffer secondary extinctions only following the extinction of all of their hosts". I disagree. They can evolve to attack new partners or they can die out before all partners are lost due to competition with other parasites and other interactions.

11. There is no discussion. The literature on rapid evolution in species interactions, coevolution, or even conservation biology is just ignored.

Finally, some results need to be better explored:

1. If coevolution is the mechanism beyond the resilience of the ecological networks explored why "Parasites were far less robust to novel host removal, which was intermediate between the best and worst-case scenarios"?

2. No information of network structure is explored. Actually, if I do suggest the authors to use assemblages instead of networks - it would be a more accurate description of the characterization of the systems they are exploring.

Finally and even more importantly some analyses/assumptions need to be double-checked to actually provide results to support/reject the claims.

3. I do not see how the results could be related to the coevolutionary process. To do so, you need to have a theoretical benchmark of how the fragility would look like in the absence of coevolution. I missed three controls - sets of simulations - in which evolution only occurs in hosts, evolution only occurs in parasites, and no evolution at all.

4. If I understood well - but I am not sure, see 9 - the novel host scenario is a random removal scenario. If so the underlying assumption is that climate change effect is random. You need to discuss this assumption in the paper.
5. Coevolution is a local process. Thus, the use of empirical, global-level networks might not provide information on coevolution. The authors need to justify these assumptions.
6. No information/analyses on how sampling biases could affect the results on empirical networks is provided but an unsupported statement "Such limitations, however, should not bias parasite extinction rates, leading instead to conservative estimates of robustness". Please provide analyses to support such strong and keystone statement.
7. Species in the numerical simulations are not species but individuals grouped by phenotypic and functional similarity. By performing this type of lumping the authors are mixing adaptive processes such as convergence with phylogenetic relatedness. This might limit the interpretation of the results and the authors need to discuss the potential limitations in the text.

Reviewer #2 (Remarks to the Author)

In this study the authors examine whether species adaptation can increase community robustness and whether such robustness can hold under external perturbation. The underline rationale is that system complexity can buffer stochastic disruption but the degree of buffering of the system depends on the type and amount of lost (host disassembly). This work establishes a valuable basis by showing how the system evolves under different scenarios.

They focus on the host-parasite network and two approaches are used, an artificial system of populations using the program Avida and real-world networks based on parasites of vertebrates from available sources. The general conclusion is that a system exposed to global host disassembly is more robust if under adaptive (status quo) than perturbed (novel and more risky) conditions. Findings also suggest that the way components of the network are removed (at random or based on their relative vulnerability) is important for network robustness. When there is a lack of information on species vulnerability then robustness appears to be comparable to a random disassembly.

The paper is interesting and well written, however, I found it difficult to understand the dynamics of the artificial model without much details on the assumptions of the system. For example, it is unclear how parasites mutate and co-evolve with the host, how transmission occurs or how hosts and parasites compete or interact in the artificial platform. There are references to previous studies and a web link but the fundamental of the model should be included in the Methods to provide a more accurate and easy-to-follow description of the system and its level of realism.

In the artificial network, parasites self-replicate and have free-living phases, this suggests to me that the system is mainly based on microparasites and their free-living conditions are known. This contrasts with the real-world network that uses macroparasites that reproduce sexually and often have intermediate hosts. Essentially, how comparable are the two systems and, more generally, are macro- and micro-parasite networks comparable in terms of species extinction and robustness of the system (which also goes back to the use of AVIA at page 9)?

Is the host species starting the disassembly important? I would expect that a host with a strong network might play a more relevant role but, then again, if it is lost the network might quickly adjust to a new status.

At page 3 lines 81 to 85, the assumption that extinction risk does not correlate with the amount of research on a particular species need to be explained more clearly in the context of model outputs. This contrasts with what said at page 9 lines 246-7 where the authors admit that there are some

biases.

Similarly, the assumption that the lack of information on complex life cycles should not bias parasite extinction rates needs to be further developed and linked more clearly to the random results and the IUCN findings. These findings become clear only in the Methods and should be brought up more clearly in the main text.

Do networks from different taxa (fishes, birds, amphibians, reptiles and mammals) behave in the same ways? In other words, is the robustness of the network comparable under different scenarios across taxa?

The conclusion should be further elaborated by including the broad implication of this study for different taxa and across spatial levels, both in the context of current stability and disturbance as a short term or long term event.

Minor comments:

Page 6 lines 175-6, the sentence needs to be corrected.

Page 7 line 178. Are hosts removed based on their increasing relative vulnerability?

Figures: it should be highlighted what is from digital and real data simulations

Reviewer #3 (Remarks to the Author)

For this work, the authors examined the robustness of digital and real-world host-parasite networks by subjecting them to different types of perturbations. I found the approach to be quite interesting, and the manuscript was well prepared. The big-picture implications—that global change could have strong effects on even robust networks—are compelling. However, given the ability to create entirely new worlds with the Avida platform, the paper left me with more questions (inspired by this work and the possibilities of the system) than answers. Although a detailed and broad analysis is not feasible in this article format, I feel as if some further exploration into these networks as well as some insights into how extinctions might be mitigated would greatly strengthen the paper.

I thought the use of real-world networks was very interesting, and nicely complemented the simulation data. I also particularly liked the empirical bias experiments, which nicely showed what can be done with imperfect information.

Much of the data presented were from artificial populations that evolved in Avida under previously-published conditions (Zaman, Fortuna, and colleagues). While I feel that this is a nice and appropriate use of Avida, I am concerned that some of the key model specifics are neither introduced nor discussed. For example, if the authors followed Zaman's model and parameters exactly, parasites consume 80% of their host's "energy", and hosts may be infected by at most one parasite. I would suspect that either of these parameters could strongly influence the resulting networks as well as the diversity, richness, robustness, etc.. Some investigation into the effects of these key parameters would provide a broader picture.

The Avida model provided the authors with data every 100 time steps. I feel that some evolutionary perspective on how these networks and their robustness, etc. change over time would be very interesting. Do systems evolve towards more fragile states?

Finally, the use of the same methods with both systems is a strength, but I also feel that not including some method of extinction that was based on host phenotype is a missed opportunity, particularly

because environmental change is likely to affect certain phenotypes. For example, removing hosts that compete Avida's AND NOT task, for example, might affect hosts with different parasite richnesses and therefore have different effects on the underlying network.

Minor Comments:

- 118: "which resources are risky might CHANGE as conditions CHANGE" could be written a bit more clearly
- 130: the phrase "complex systems" is used a lot, which could have different connotations for readers. Would "complex ECOSystems" be more appropriate?
- 152: 400 parasites are introduced. I realize that this is following with previous work, but I was bothered that this number wasn't motivated.
- 155: consider adding "of simulations" after "at the end".
- 1157: add comma after "i.e."
- 1191: "ecological runs" is used, but I'm not sure all readers would understand what this means. Perhaps considering introducing this term when it is used.
- 1199: although the complete data set is very large, would it be possible to create a smaller data set that contains only the data from the variables in question? I would imagine that these data would be open to lots of interesting analysis. Avida writes a lot of data, most of which is not needed in all instances.
- Figure 2a: I'd like to see the variance in these data
- Figure 2b: I personally don't like specific types of plots used as the subject in figure captions. Instead of "box plots", describe what is actually shown. Here, it's the Robustness.
- Figure 4: Re-describe what the different components of the box plots represent (since different software can use different values).

Reviewers' Comments:

Reviewer #2 (Remarks to the Author)

This is a well improved new version of the manuscript; the approach and methods have been more clearly described and the results are more cohesive and consistent with the goals and general rationale of the work.

Minor comments

Page 3 line 58-60. Reference to fig 1a and 1b should be the other way round
Page 4/5 line 109-111 and again 117-119 are very much the same sentence; perhaps adjust it.
Table 1S and 2S are not straightforward to understand and need a better legend. Also, shouldn't table 1S present these results for the different scenarios rather than averaging across them?

Reviewer #3 (Remarks to the Author)

Substantial changes have been made to this manuscript, and I applaud the authors' careful consideration of my and the other reviewers' comments. The results are now more robust, and I would imagine that the additions in text, experiments, and results would address a number of issues that readers might commonly encounter. I certainly enjoyed learning more about this process through their new results.

As in my previous review, I believe that this work is interesting, novel, and provides unique insights into a pressing problem of global concern. The paper is well written, and the included figures appropriately and effectively portray the results.

In response to reviewer comments, the authors have examined how several additional model parameters affect their results through randomization of these parameters (I assume that values were chosen at the onset of simulations and remained fixed throughout individual simulations—a detail that should be made more explicit). I personally would have preferred simulations where values were swept for single key parameters, thus directly probing their effects, but I realize how this would increase the number of simulations required quite dramatically. While the authors do show that their results were not sensitive to these new perturbations, I feel like my grasp for the system and my understanding of the results is a bit weaker given all the variables at play (and possible interactions among them).

In the Avida model, infection by multiple parasites is not possible. I don't believe that this a show-stopper, but I do believe that it should be mentioned, perhaps in the expanded discussion, so that the reader might consider what effect multiplicity could have on this process. To further another reviewer comment, I think the paper could also benefit from a (very) brief description of parasite transmission.

As for the experimental design, the results are mostly descriptive of what the data show. It seems like there's plenty of room for alternate hypotheses, but they are not introduced or tested.

Some specifics:

Abstract:

- "and IF such robustness" -> "and WHETHER such robustness"
- "REAL host-parasite". This is probably ok, but I feel like REAL diminishes the contribution of the model. NATURAL?

L26: I'm not familiar with the condor louse example, so explain what the example actually is. Did the louse go extinct following the condor's extinction?

L52: "for EACH OF 100 evolution simulations"?

L56: "varied in their virulence" - "virulence" seems like an over-loaded term here. Does this mean that they varied both in how many CPU cycles they took from their hosts AND their host preference?

Figure 1:

- L58, 59: "1A" and "1B" used instead of "1a", "1b"
- L58, 59, 166: "1A" used to refer to the plot in "1B", and vice versa
- Caption: Generalism (b) discussed first. Robustness is not described. Also needs subplot labels.
- Text discusses specialization, while plot shows generalization. It would probably be better to describe what exactly is being shown instead of its inverse.

L81: Showing an example network as a Figure or Supplemental Figure would help readers visualize the networks and probably also the disassembly process.

L99: "or IF we removed hosts" -> "WHETHER"

L110, 118: DISTRIBUTIONS. Distributions of what, space?

L128: "IT was still higher" - Does the IT refer to "assemblage robustness". Since this is a result, I think it should be completely clear.

L175: "Fig S1b" does not exist.

L182: While interesting, I don't think the computational requirements for the work is necessary and will lose meaning quickly due to advances in computing

L189: "randomized a setting already explored" - which setting exactly? Population size?

L193: "a situation in which the parasiteS steal all the CPU CYCLES OF its host"

L195: "MAIN nine" - Avida has alternate task sets (drawing a blank on references, sorry), so MAIN seems inappropriate even though these nine are most commonly used. CANONICAL?

L197: randomly chosen at the onset, but fixed for the duration of a simulation?

L256: Table S1 doesn't appear to describe networks

L291: Citation should probably be given for the IUCN redlist

Figure 2: Not obvious what "position" is, so this figure could be difficult to approach for many.

Figure 5 + Figure 6: These could be paired just like Figures 3a and 3b

Figure 6:

- "Robustness MEANS" -> DESCRIBES/REPRESENTS/...
- 95% CI lines: Would asterisks denoting significance at some level be more familiar to readers?
- I like the fish, but since it's the only such image in this figure, it seems a little out of place

Figure S1: The linear model that's described should be shown

Figure S2: Having a reference in a Figure title seems strange

Reviewer #1 (Remarks to the Author):

I read with interest the paper by Strona and Lafferty. The simulations using Avida were a very interesting approach to the problem on how coevolution affects the fragility of ecological networks. Having said that I am afraid there are serious issues with the current draft.

First, there is no assessment of the generality of the results.

1. For example, are hundred simulations enough? The authors did not provide any evidence this is a good number of simulations.

Authors: Actually, 100 simulations was overkill. The ideas behind Avida are close to R. Lenski's long-term evolution experiment on *E. coli*. that has been at the basis of fundamental discoveries in various evolutionary topics. Such experiments provided robust (i.e. replicable) results relying on 'just' 12 evolutionary lines. We based our sample size on previous experiments with Avida that found 100 simulations gave high consistency across communities. 100 simulations turned out to be more than enough to give the statistical power we needed to contrast the large effects we saw. As an aside, with the new simulations, we now have done twice the simulations compared with other papers and they match our independent 100 simulations replicated with a new set of co-evolved communities (see next response below).

2. More importantly, there is no sensitivity analysis on the parameter values used or on modeling decisions (e.g., 400 parasites copies, 2000 generations, and so on).

3. As a consequence - and if I understood right - all the results reported derive from a single combination of parameters and model decisions (which are, of course, implicit parameters of the model). If so, the results are actually preliminary.

4. As a consequence of 1-3 it is impossible to make any assessment of the generality of the results and at it stands the work is a promising set of results, but it is not a solid contribution one would expect in a top journal as Nature Communications.

Authors: To increase the generality of the results, we replicated all our analyses using a different experimental design where model parameters were randomly varied. In particular, in the new set of experiments we permitted random variability in the size of the Avida world, in the number and the timing of parasite injections, and, most notably, in the number and quantity of available resources and of consumption products. This last aspect permitted us to create very different and complex scenarios. In addition, instead of using a fixed virulence for parasites we have allowed parasites to evolve their own virulence through co-evolutionary dynamics. In short, the results were not sensitive to these permutations. The new analyses produced results consistent with those we reported in the first draft, which provides strong evidence of the generality of the patterns.

Second, I could not evaluate the quality of the work presented because I simply could not follow several aspects of the draft. Examples include but are not limited to:

5. I assume the r_s values are the Spearman's rho but if so how it tell us about specialization? Is about how specialization varies with host survival? If so it is not clear. Line 59.

Authors: We have better clarified the relationships between parasite specialization/richness and host extinction risk.

6. Figure 1: how can p-values be smaller than zero?

Authors: We have corrected the figure, modifying the value to $p < 0.001$

7. If you are "halting mutation and letting host species outcompete each other in the environmental conditions they had evolved" you are not evaluating the parasitism effect but the consequences of parasites to competition. Is this right?

Authors: We see that this might be confusing and have worked to clarify it. Our simulation works in two phases. The coevolution phase creates coevolved communities, including coevolved parasite-host interactions. The assessment phase allows us to measure the vulnerability of each host to extinction in a sort of 'arena' experiment where the stronger competitors drove weaker ones to extinction. We left parasites in the system during assessment in order to preserve their potential effect on host competition dynamics.

8. What is the meaning of "parasite robustness"? I do not follow why "removing hosts in decreasing order with parasite richness" is "the best-case scenario for parasite robustness".

Authors: Parasite robustness and food web robustness are types of network robustness. We use the term parasite robustness (in place of network robustness) so that readers don't confuse it with food web robustness. Parasite robustness quantifies how slowly parasite species richness decline as hosts (and only hosts) are removed in sequence. Food web robustness, on the other hand, refers to how the number of species in a food web declines as species are removed at random. Food web robustness is more appropriate to food webs where all consumers (but not basal taxa) can suffer secondary extinctions as species are removed in sequence. We have defined and clarified our use of parasite robustness to distinguish it from food web robustness and the broader network robustness.

As for worst and best case scenarios, we have clarified that parasites are driven extinct if we remove their hosts, but not if we remove their hosts' competitors. The worst-case scenario for the parasite community, therefore, is to first remove hosts that support the most parasites. The best-case scenario for parasites is to first remove hosts that don't have parasites. These are obviously an important point to clarify, so we have now provided a clearer definition of 'robustness' and we have better explained best and worst case scenarios.

9. What is the meaning of "approximated as the average of 100 random orderings"?

Authors: we have now better clarified (in the Methods) how we computed robustness by averaging the results of 100 replicates where the positions of ties in the host removal sequences were randomized.

10. parasites suffer secondary extinctions only following the extinction of all of their hosts". I disagree. They can evolve to attack new partners or they can die out before all partners are lost due to competition with other parasites and other interactions.

Authors: The reviewer is correct. The confusion arose because we were referring to the specifics of the simulation, not general aspects of parasite extinction. To avoid confusion for the reader, we have now removed this statement, and better clarified the reasons why we focused on host-parasite networks (and particularly the self-contained response of parasites to host extinctions).

11. There is no discussion. The literature on rapid evolution in species interactions, coevolution, or even conservation biology is just ignored.

Authors: we have extended the discussion, including references to the topics highlighted by Reviewer 1.

Finally, some results need to be better explored:

1. If coevolution is the mechanism beyond the resilience of the ecological networks explored why "Parasites were far less robust to novel host removal, which was intermediate between the best and worst-case scenarios"?

Authors: we have now better clarified that the mechanisms responsible for this result lay in the patterns shown in Figure 2. Specifically, in the 'natural' (observed) sequences of host extinctions, hosts having many parasites and/or specialized parasites tend to stay in the system longer than hosts having few, generalist parasites. In the best-case scenario, we forced hosts having many parasites to stay in the system as long as possible, while in the worst case scenario we reversed the sequence, favoring the persistence of hosts having few parasites. In a random sequence of host extinction, we attributed to hosts having many parasites with the same probability of going extinct than that of hosts having many parasites. This led the random case to have a robustness intermediate between the best and the worst case scenario.

2. No information of network structure is explored. Actually, if I do suggest the authors to use assemblages instead of networks - it would be a more accurate description of the characterization of the systems they are exploring.

Authors: Networks are assemblages and assemblages can be networks. Here is why we use *Network* when referring to hosts and parasites instead of *Assemblage*. To describe an Assemblage one uses lists of co-occurring species, without information on interactions and dependencies. A bipartite network, on the other hand, contains information on links as a topology and that information is necessary to estimate robustness to secondary extinction. However, we now use the term Assemblage when referring to just the hosts or just the parasites. We have now investigated network structure, measuring, amongst other things, nestedness, segregation, and modularity, with the additional purpose of showing how the random parametrization of the Avida setting permitted us to create and explore very different scenarios. We have added a figure reporting the frequency distribution of nestedness and modularity among networks, and a table summarizing the main network properties. In addition, we have investigated the relationships between network structure (and Avida settings) and robustness (see table S1).

Finally and even more importantly some analyses/assumptions need to be double-checked to actually provide results to support/reject the claims.

3. I do not see how the results could be related to the coevolutionary process. To do so, you need to have a theoretical benchmark of how the fragility would look like in the absence of coevolution. I missed three controls - sets of simulations - in which evolution only occurs in hosts, evolution only occurs in parasites, and no evolution at all.

Authors: The reviewer raises an important distinction that we have clarified. We cannot distinguish the role of coevolution from evolution because none of the three controls the reviewer suggests would permit the emergence of host-parasite networks in Avida. First, a lack of diversification would limit network richness. Also, preventing parasites from evolving would result in parasite extinction (for example, newly evolved parasite-free hosts would rapidly outcompete parasitized hosts, driving parasites to extinction). That said, in Avida parasites evolve in response to hosts and hosts evolve in response to parasites – so this is a coevolved network (see papers by Zaman). Still, given the important meaning of this term, we have looked to avoid confusion by not focusing on coevolution as a driving factor, focusing instead on the historically robust networks as the issue we are testing (while still explaining to the reader how these are formed by coevolution). Note that we took coevolution out of the title to reduce confusion by readers that we were somehow testing coevolution as a factor.

4. If I understood well - but I am not sure, see 9 - the novel host scenario is a random removal scenario. If so the underlying assumption is that climate change effect is random. You need to discuss this assumption in the paper.

Authors: we have now better clarified that we are not modeling climate change in Avida, which would have non-random effects on species. Randomness is just the way that we generate change without information on what that change would look like, highlighting how this indicates simply that species vulnerability differs from that expected under the conditions in which the system has evolved. Our findings on the real host-parasite datasets strongly support this idea, showing how the current species vulnerability as assessed by IUCN criteria is statistically indistinguishable from a random scenario. I.e., the details of the change matter, but it is mostly about the extent that the future differs from the past. To further avoid confusion on this matter we have taken the term global change out of the title. We are more concerned with change, than the source of change.

5. Coevolution is a local process. Thus, the use of empirical, global-level networks might not provide information on coevolution. The authors need to justify these assumptions.

Authors: We have clarified in the text that the global host-parasite databases have different assumptions than the local processes we model with Avida. As mentioned, we use Avida to generate stable networks based on coevolution. The global networks are also presumably stable, though the processes that created them are different. Regardless, both approaches deal with the relationships between resource dependability, and parasite richness and specificity. Local environmental factors and biogeographical mechanisms may clearly affect species dependability, but do not have a direct effect on the relationships between this and parasite richness/specialization. We do not use coevolution when describing the empirical networks.

6. No information/analyses on how sampling biases could affect the results on empirical networks is provided but an unsupported statement "Such limitations, however, should not bias parasite extinction rates, leading instead to conservative estimates of robustness". Please provide analyses to support such strong and keystone statement.

Authors: We now provide an extensive analysis and discussion of the potential effects of sampling bias (and complex life cycles, and various other factors). We found no evidence that our assumptions were greatly violated, at least to the extent that they would influence the qualitative findings (apart from making them a little conservative).

7. Species in the numerical simulations are not species but individuals grouped by phenotypic and functional similarity. By performing this type of lumping the authors are mixing adaptive processes such as convergence with phylogenetic relatedness. This might limit the interpretation of the results and the authors need to discuss the potential limitations in the text.

Authors: We have clarified that the main scope of our analysis was to investigate the robustness of ecological networks to global perturbations. To this purpose, focusing on functional categories such as those provided by Avidian phenotypes is more informative than focusing on genotypes (see Zaman et al. 2013). Nevertheless, to investigate potential issues deriving from our choice of focusing on phenotypes we have replicated the disassembling analyses using genotypes, finding very consistent results (reported as Supplementary information).

Reviewer #2 (Remarks to the Author):

In this study the authors examine whether species adaptation can increase community robustness and whether such robustness can hold under external perturbation. The underline rationale is that system complexity can buffer stochastic disruption but the degree of buffering of the system depends on the type and amount of lost (host disassembly). This work establishes a valuable basis by showing how the system evolves under different scenarios.

They focus on the host-parasite network and two approaches are used, an artificial system of populations using the program Avida and real-world networks based on parasites of vertebrates from available sources. The general conclusion is that a system exposed to global host disassembly is more robust if under adaptive (status quo) than perturbed (novel and more risky) conditions. Findings also suggest that the way components of the network are removed (at random or based on their relative vulnerability) is important for network robustness. When there is a lack of information on species vulnerability then robustness appears to be comparable to a random disassembly.

The paper is interesting and well written, however, I found it difficult to understand the dynamics of the artificial model without much details on the assumptions of the system. For example, it is unclear how parasites mutate and co-evolve with the host, how transmission occurs or how hosts and parasites

compete or interact in the artificial platform. There are references to previous studies and a web link but the fundamental of the model should be included in the Methods to provide a more accurate and easy-to-follow description of the system and its level of realism.

Authors: We have extended the Methods, including more details about the Avida system and our experimental setup.

In the artificial network, parasites self-replicate and have free-living phases, this suggests to me that the system is mainly based on microparasites and their free-living conditions are known. This contrasts with the real-world network that uses macroparasites that reproduce sexually and often have intermediate hosts. Essentially, how comparable are the two systems and, more generally, are macro- and micro-parasite networks comparable in terms of species extinction and robustness of the system (which also goes back to the use of AVIA at page 9)?

Authors: We have now specified that in Avida parasites do not have free living phases (see Methods). Regardless, there are no intermediate hosts in Avida, so the reviewer is correct that this is more a microparasite framework that differs from many of the real-world macroparasites in our database. To clarify this difference, we have added a reference to Rudolf and Lafferty's work showing how complex macroparasite life cycles reduce parasite robustness.

Is the host species starting the disassembly important? I would expect that a host with a strong network might play a more relevant role but, then again, if it is lost the network might quickly adjust to a new status.

In some cases, disassemblies can be sensitive to particular species. However, these networks are large enough and parasites are broadly enough distributed that the first host to leave does not have a large effect on the robustness of the parasite community (though it could have a large effect on parasites specific to it). Because we don't focus on single parasite species, this is not an issue for our results. Also, our randomized disassemblies are composite averages of 100 sequences, further reducing the importance of one species.

At page 3 lines 81 to 85, the assumption that extinction risk does not correlate with the amount of research on a particular species needs to be explained more clearly in the context of model outputs. This contrasts with what is said at page 9 lines 246-7 where the authors admit that there are some biases. Similarly, the assumption that the lack of information on complex life cycles should not bias parasite extinction rates needs to be further developed and linked more clearly to the random results and the IUCN findings. These findings become clear only in the Methods and should be brought up more clearly in the main text.

Authors: We have now discussed in more detail how sampling bias can affect our results, and why it makes them, in fact, even more conservative. Furthermore we have discussed the potential effect of complex life cycles on our results, and how this makes our findings conservative (i.e. real networks are likely even less robust than we show).

Do networks from different taxa (fishes, birds, amphibians, reptiles and mammals) behave in the same ways? In other words, is the robustness of the network comparable under different scenarios across taxa?

Authors: we have now compared the robustness of networks from different taxa (both for host and parasite taxa) finding no significant effect (ANOVA $p=0.517$ for host taxa and $p=0.723$ for parasite taxa), confirming that the mechanisms we have identified (and hence the risk for the future) are much general. We have reported these results in the MS.

The conclusion should be further elaborated by including the broad implication of this study for different taxa and across spatial levels, both in the context of current stability and disturbance as a short term or long term event.

We have added this to the Discussion.

Minor comments:

Page 6 lines 175-6, the sentence needs to be corrected.

Authors: we have corrected, distinguishing the best from the worst case scenario.

Page 7 line 178. Are hosts removed based on their increasing relative vulnerability?

Authors: we have corrected the sentence by specifying that we removed host in decreasing order of vulnerability (i.e. we removed first the most vulnerable hosts).

Figures: it should be highlighted what is from digital and real data simulations

Authors: We have better clarified in the legends which figures refer to real data and which ones refer to simulations.

Reviewer #3 (Remarks to the Author):

For this work, the authors examined the robustness of digital and real-world host-parasite networks by subjecting them to different types of perturbations. I found the approach to be quite interesting, and the manuscript was well prepared. The big-picture implications-that global change could have strong effects on even robust networks-are compelling. However, given the ability to create entirely new worlds with the Avida platform, the paper left me with more questions (inspired by this work and the possibilities of the system) than answers. Although a detailed and broad analysis is not feasible in this article format, I feel as if some further exploration into these networks as well as some insights into how extinctions might be mitigated would greatly strengthen the paper.

I thought the use of real-world networks was very interesting, and nicely complemented the simulation data. I also particularly liked the empirical bias experiments, which nicely showed what can be done with imperfect information.

Much of the data presented were from artificial populations that evolved in Avida under previously-published conditions (Zaman, Fortuna, and colleagues). While I feel that this is a nice and appropriate use of Avida, I am concerned that some of the key model specifics are neither introduced nor discussed.

For example, if the authors followed Zaman's model and parameters exactly, parasites consume 80% of their host's "energy", and hosts may be infected by at most one parasite. I would suspect that either of these parameters could strongly influence the resulting networks as well as the diversity, richness, robustness, etc.. Some investigation into the effects of these key parameters would provide a broader picture.

Authors: We have replicated all the experiments by randomizing several model parameters. With regards to the specific points raised by Reviewer 3, in the new set of experiments we have let parasite virulence evolve throughout the simulations. The limitation of having one parasite per host, however, cannot be overcome in the current version of Avida (Zaman, personal communication). Despite this new broader experimental design, we found very consistent results with those obtained in our first set of experiments, which provides strong support to our conclusions.

The Avida model provided the authors with data every 100 time steps. I feel that some evolutionary perspective on how these networks and their robustness, etc. change over time would be very interesting. Do systems evolve towards more fragile states?

Authors: Wow. Great question (we hadn't thought of that). We have now performed two additional tests to investigate the evolution of robustness through time. First we have compared the robustness of networks at different stages of maturity in the context of our main experiment (in which we stopped evolution at random between 100000 and 500000 generations). Since we found no relationship between robustness and maturity, we hypothesized that the networks become robust before 100000 generations. Thus, we picked up a random simulation, and we investigated the evolution of robustness in the host/parasite networks from the beginning of the co-evolutionary process (10000 generations) to the maturity stage (100000). As we discuss, robustness does change over time, with some interesting implications, especially in the face of random change.

Finally, the use of the same methods with both systems is a strength, but I also feel that not including some method of extinction that was based on host phenotype is a missed opportunity, particularly because environmental change is likely to affect certain phenotypes. For example, removing hosts that compete Avida's AND NOT task, for example, might affect hosts with different parasite richnesses and therefore have different effects on the underlying network.

Authors: Following the Reviewer's suggestion we have now analyzed the effect of phenotypes on robustness, by performing removal experiments where we focused on individual tasks

Minor Comments:

- 118: "which resources are risky might CHANGE as conditions CHANGE" could be written a bit more clearly

Authors: we have modified the sentence.

- 130: the phrase "complex systems" is used a lot, which could have different connotations for readers. Would "complex ECOSystems" be more appropriate?

Authors: we have now replaced the word 'system' with a more accurate expression ('ecosystem', 'assemblage', or network) depending on the specific context.

- l52: 400 parasites are introduced. I realize that this is following with previous work, but I was bothered that this number wasn't motivated.

Authors: we have now randomized the injection time (and the amount of injected parasites).

- l55: consider adding "of simulations" after "at the end".

Authors: done

- l157: add comma after "i.e."

Authors: done (also in other parts of the MS).

- l191: "ecological runs" is used, but I'm not sure all readers would understand what this means. Perhaps considering introducing this term when it is used.

Authors: we have modified the text making explicit that we meant the second phase of the experiment (i.e. after we stopped mutations).

- l199: although the complete data set is very large, would it be possible to create a smaller data set that contains only the data from the variables in question? I would imagine that these data would be open to lots of interesting analysis. Avida writes a lot of data, most of which is not needed in all instances.

Authors: We are now providing the co-evolved networks and the host extinction sequences as Supplementary Data.

- Figure 2a: I'd like to see the variance in these data

Authors: We have added 95% confidence intervals to the figure.

- Figure 2b: I personally don't like specific types of plots used as the subject in figure captions. Instead of "box plots", describe what is actually shown. Here, it's the Robustness.

Authors: we have modified the legend according to this suggestion.

- Figure 4: Re-describe what the different components of the box plots represent (since different software can use different values).

Authors: we have modified the legend according to this suggestion.

Reviewer #2 (Remarks to the Author):

This is a well improved new version of the manuscript; the approach and methods have been more clearly described and the results are more cohesive and consistent with the goals and general rationale of the work.

Minor comments

Page 3 line 58-60. Reference to fig 1a and 1b should be the other way round

Authors: corrected.

Page 4/5 line 109-111 and again 117-119 are very much the same sentence; perhaps adjust it.

Authors: we have removed the duplicated sentence.

Table 1S and 2S are not straightforward to understand and need a better legend. Also, shouldn't table 1S present these results for the different scenarios rather than averaging across them?

Authors: We have improved the legends of the two tables. The aim of Table S1 was that of showing that we explored a broad range of model parameters, and that this led to differences in the co-evolved communities. We therefore think that showing the summary statistics (including not only the average but also the variability of the data) conveys the idea more straightforwardly than reporting the details for all the 100 simulations.

Reviewer #3 (Remarks to the Author):

Substantial changes have been made to this manuscript, and I applaud the authors' careful consideration of my and the other reviewers' comments. The results are now more robust, and I would imagine that the additions in text, experiments, and results would address a number of issues that readers might commonly encounter. I certainly enjoyed learning more about this process through their new results. As in my previous review, I believe that this work is interesting, novel, and provides unique insights into a pressing problem of global concern. The paper is well written, and the included figures appropriately and effectively portray the results.

In response to reviewer comments, the authors have examined how several additional model parameters affect their results through randomization of these parameters (I assume that values were chosen at the onset of simulations and remained fixed throughout individual simulations—a detail that should be made more explicit). I personally would have preferred simulations where values were swept for single key parameters, thus directly probing their effects, but I realize how this would increase the number of simulations required quite dramatically. While the authors do show that their results were not sensitive to these new perturbations, I feel like my grasp for the system and my understanding of the results is a bit weaker given all the variables at play (and possible interactions among them).

Authors: as the Reviewer highlighted, performing separate sensitivity analyses on all model parameter would have been computationally prohibitive. Nevertheless, the aim of parameter randomization was mostly testing the generality of our results. Individual variables may have specific effects on the structure of the co-evolved host/parasite networks, but the consistency of our results

strongly suggests that they do not affect our key findings (and particularly, that networks are robust to stable conditions but not to changing ones).

In the Avida model, infection by multiple parasites is not possible. I don't believe that this a show-stopper, but I do believe that it should be mentioned, perhaps in the expanded discussion, so that the reader might consider what effect multiplicity could have on this process. To further another reviewer comment, I think the paper could also benefit from a (very) brief description of parasite transmission.

Authors: information about transmission strategy and the limit of one parasite per host is provided in lines 243-247: "Parasites in Avida are similar to free-living species in terms of structure and evolutionary processes (i.e., mutation type and rate). However, parasites could not survive outside a host. Thus, when a parasite reproduces, its offspring try to infect a nearby host (like a directly transmitted, single-host microparasite). The host is susceptible only if it is uninfected, and the parasite can do at least one of its tasks."

As for the experimental design, the results are mostly descriptive of what the data show. It seems like there's plenty of room for alternate hypotheses, but they are not introduced or tested.

Authors: This is laid out in detail in the section on Sensitivity analyses.

Some specifics:

Abstract:

- "and IF such robustness" -> "and WHETHER such robustness"

Authors: corrected.

- "REAL host-parasite". This is probably ok, but I feel like REAL diminishes the contribution of the model. NATURAL?

Authors: we modified REAL to EMPIRICAL.

L26: I'm not familiar with the condor louse example, so explain what the example actually is. Did the louse go extinct following the condor's extinction?

Authors: we now make explicit that the louse went extinct.

L52: "for EACH OF 100 evolution simulations"?

Authors: corrected.

L56: "varied in their virulence" - "virulence" seems like an over-loaded term here. Does this mean that they varied both in how many CPU cycles they took from their hosts AND their host preference?

Authors: we now specify that “virulence” refers only to the amount of CPU a parasite steals from its host.

Figure 1:

- L58, 59: "1A" and "1B" used instead of "1a", "1b"

Authors: corrected.

- L58, 59, 166: "1A" used to refer to the plot in "1B", and vice versa

Authors: corrected.

- Caption: Generalism (b) discussed first. Robustness is not described. Also needs subplot labels.

Authors: corrected.

- Text discusses specialization, while plot shows generalization. It would probably be better to describe what exactly is being shown instead of its inverse.

Authors: we have modified both fig. 1 and fig. S1 to show specialization instead of generalism.

L81: Showing an example network as a Figure or Supplemental Figure would help readers visualize the networks and probably also the disassembly process.

Authors: we suspect that for some reasons the Reviewer has missed Movie S1. We think that this animation, more than a static figure, can provide a clear idea of the networks and of the disassembly process.

L99: "or IF we removed hosts" -> "WHETHER"

Authors: corrected.

L110, 118: DISTRIBUTIONS. Distributions of what, space?

Authors: we now clarify that we refer to geographical distribution.

L128: "IT was still higher" - Does the IT refer to "assemblage robustness". Since this is a result, I think it should be completely clear.

Authors: we now clarify that we refer to assemblage robustness.

L175: "Fig S1b" does not exist.

Authors: corrected (to Fig. S1).

L182: While interesting, I don't think the computational requirements for the work is necessary and will lose meaning quickly due to advances in computing

Authors: the statement was meant to emphasize that performing broad experiments on Avida requires the use of high performing computational facilities. However, we have removed the statement.

L189: "randomized a setting already explored" - which setting exactly? Population size?

Authors: we now specify the randomized parameters:

“To make our results as general as possible, we randomized several parameters (carrying capacity, parasite virulence, resource availability, injection timing and amount of ancestral parasites) of a setting already explored in host-parasite co-evolution experiments”

L193: "a situation in which the parasiteS steal all the CPU CYCLES OF its host"

Authors: corrected

L195: "MAIN nine" - Avida has alternate task sets (drawing a blank on references, sorry), so MAIN seems inappropriate even though these nine are most commonly used. CANONICAL?

Authors: modified to “canonical”

L197: randomly chosen at the onset, but fixed for the duration of a simulation?

Authors: we have rephrased to make clear that the input-output ratio is set at the beginning and fixed throughout the simulation.

L256: Table S1 doesn't appear to describe networks

Authors: we have removed the reference to Table 1

L291: Citation should probably be given for the IUCN redlist

Authors: we have added a hyperlink to IUCN website

Figure 2: Not obvious what "position" is, so this figure could be difficult to approach for many.

Authors: we now specify that we are referring to the “ordinal position in the extinction sequence”.

Figure 5 + Figure 6: These could be paired just like Figures 3a and 3b

Authors: we have deliberately avoided merging the two figures, because, as opposed to Figs. 3a and 3b, Figs. 5 and 6 refer to different datasets. In particular, Fig. 5 refers to fish only, while Fig. 6 refers to

fish plus all the terrestrial vertebrates. Merging the two figures separated would give the wrong impression that the lines in Fig. 5 are the average of the disassembling curves used to estimate the AUCs shown in the boxplots of Fig. 6.

Figure 6:

- "Robustness MEANS" -> DESCRIBES/REPRESENTS/...

Authors: corrected

- 95% CI lines: Would asterisks denoting significance at some level be more familiar to readers?

Authors: Because we are comparing 4 items to each other (rather than to a reference point) an asterix would not indicate what were are trying to show.

- I like the fish, but since it's the only such image in this figure, it seems a little out of place

Authors: we have removed the fish.

Figure S1: The linear model that's described should be shown

Authors: we now show the regression line

Figure S2: Having a reference in a Figure title seems strange

Authors: We have removed the reference from the title.